# *"On-water"* photosensitization enables redox neutral acylation and alkylation of quinones

Tanumoy Mandal[1,2], Rohan Sharma ®[1,2], Enrique Mendez-Vega[1], Julia Rehbein[1] & Burkhard König ®[1] ✉

Water-oil interfaces exhibit a high cohesive energy density and form a supramolecular hydrogen-bonding network that supports organic reactions in various ways. Here, we introduce the redox-neutral photo-Friedel-Crafts acylation and alkylation of quinones at the aqueous-organic interface. Spectroscopic evidence and computational studies indicate extensive hydrogen bonding at the water surface, enabling the stabilization of the quinone's photo-excited state and thus reducing its excited state energy, while increasing the excited state energy of the photosensitizer Eosin Y. This combined spectroscopic behavior allows photosensitization of quinone under visible light at the oil-water interface and facilitates the desired transformation. Mechanistic studies reveal that C–C bond formation occurs via hydrogen atom transfer (HAT) from the aldehyde or alkyl reactant to the quinone, following an overall redox-neutral route, with concurrent radical recombination, efficiently producing 2-functionalized quinols. The versatility of the method is demonstrated with aromatic and aliphatic aldehydes, including natural and synthetic drug molecules, as well as ethers, thioethers, alkanes, silanes, and amines, which act as acylating or alkylating agents. The reactions have also been scaled up, and the acylated quinol products have been further functionalized to showcase their synthetic potential.

Water is the most environmentally benign and biocompatible solvent, serving as the medium for biochemical reactions in living organisms[1–3]. The combined effect of hydrogen bonding, hydrophobic interactions, and high cohesive energy values (Fig. 1A. (i)) of water enhances the binding of substrates to enzymes, assists vesicle formation, and facilitates ligand or drug binding to receptors[4–9]. Therefore, in the field of organic synthesis, there is an increasing demand for the methods that use water as the reaction medium[5,10–12]. To demonstrate this, the preliminary research on Diels-Alder reactions by Breslow and later by Sharpless emphasized the unique reactivity at the water-organic interface[13–15]. The immiscible reactants stay agitated in the aqueous suspension, which enhances the reaction rate, a phenomenon known as the "on-water effect." Further studies revealed that the reaction rates of classical [2 + 2] and [4 + 2] cycloaddition reactions could be significantly accelerated on the water surface (Fig. 1A. (ii))[9,16,17]. Additional research found that a heterogeneous micellar environment can increase the local concentrations of reactants, and the combined effect of hydrogen bonding (H-bonding) and trans-hydrogen bonding can increase the nucleophilicity and leaving ability of certain substrates (Fig. 1A. (iii)) to promote aromatic nucleophilic substitution reactions ($S_NAr$)[18–21].

In recent years, photoredox catalysis has become a promising tool in organic synthesis due to operational simplicity and precise reactivity[10–12,18]. Subsequently, the utilization of water as a solvent in photochemical reactions has received considerable interest among synthetic chemists[13–15]. In this realm, our group's recent contribution, demonstrated that the oil-water interface facilitates

[1]Fakultät für Chemie und Pharmazie, Universität Regensburg, Regensburg, Germany. [2]These authors contributed equally: Tanumoy Mandal, Rohan Sharma. ✉e-mail: burkhard.koenig@ur.de

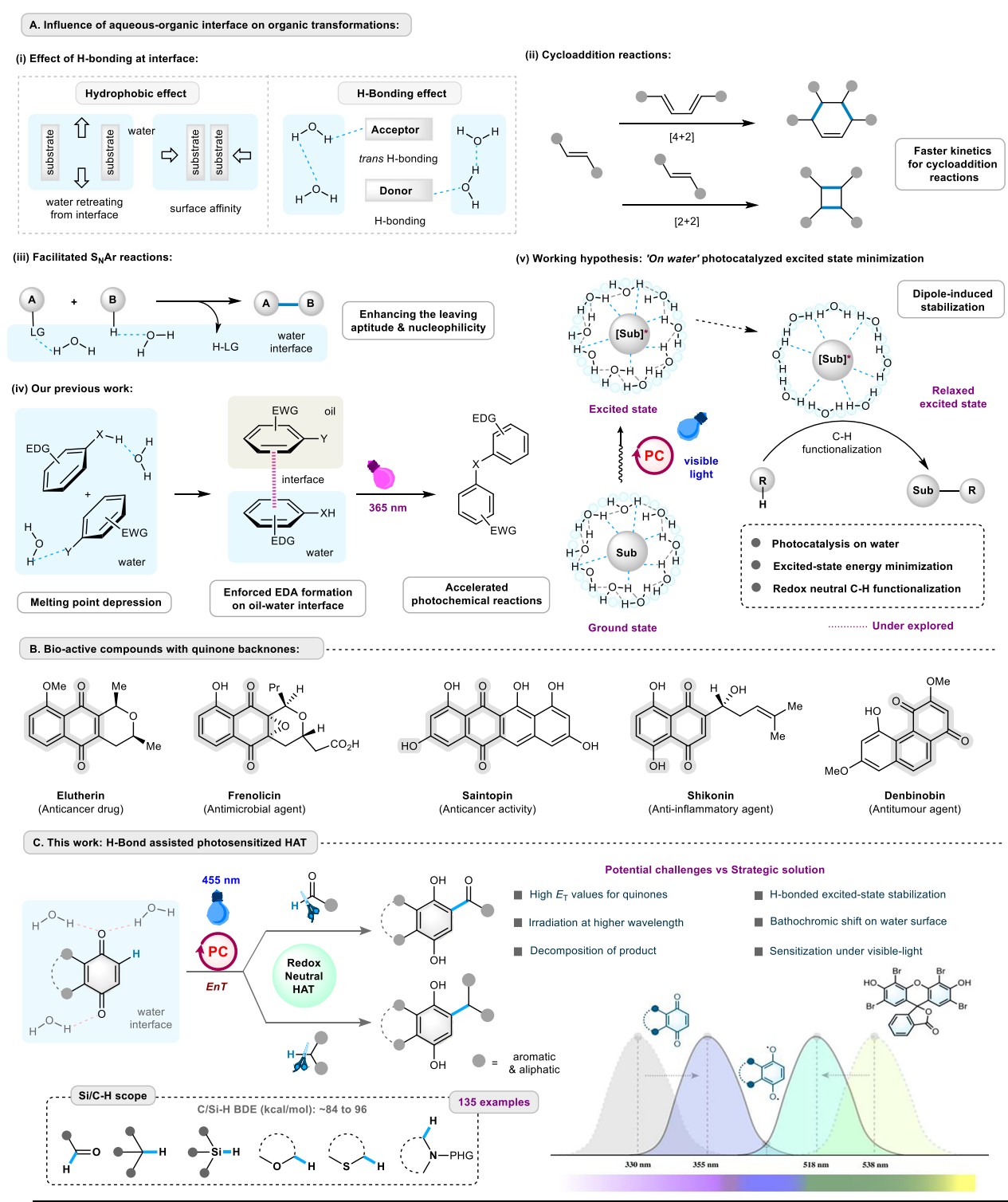

**Fig. 1 | General overview of H-bond-assisted organic transformations at the aqueous-organic interface. A** influence of hydrogen bonding (H-bonding) on organic transformations and working hypothesis; **B** bio-active molecules containing quinone backbones; **C** synthetic strategies and outcome for the present work; EDG: electron donating group, EWG: electron withdrawing group, Sub: substrate, H-bonding: hydrogen bonding, *En*T: energy transfer, $E_T$: triplet energy, nm: nanometer.

the formation of an enforced electron donor-acceptor (EDA) complex in the ground state between an electron-rich and an electron-poor molecule through melting point depression[16,17]. Further photoexcitation of such an EDA complex afforded various cross-coupling reactions with accelerated reaction kinetics (Fig. 1A.

(iv)). Nevertheless, photocatalytic reactions involving excited-state stabilization via supramolecular H-bonding on the water surface are still underexplored[19–21]. Fundamental studies on solvent polarization effect reveal that the photo-excited state of polar organic substances can be stabilized through H-bonding and dipole

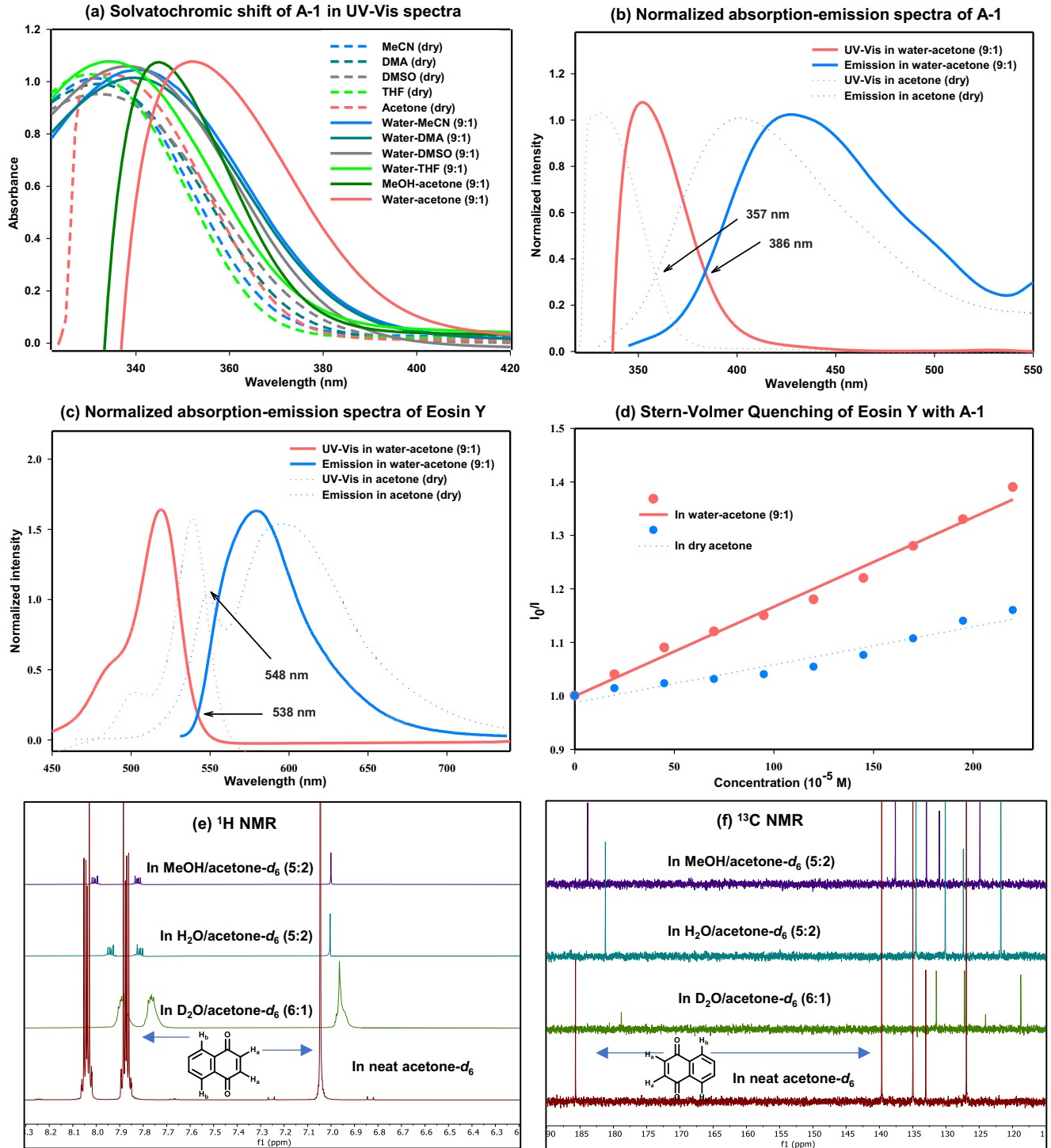

**Fig. 2 | Spectroscopic investigation. a** UV-Vis spectra of **A-1** in different anhydrous solvents and solvent/water (1:9) mixture; **b** Normalized absorption-emission spectra of **A-1** in dry acetone and water/acetone mixture (9:1); **c** Normalized absorption-emission spectra of Eosin-Y in dry acetone and water/acetone (9:1) mixture; **d** Stern-Volmer quenching plot of Eosin-Y with **A-1** in dry acetone and water/acetone mixture (9:1); **e** ¹H NMR of **A-1** in different solvent mixtures; **f** ¹³C NMR of **A-1** in different solvent mixtures; MeCN: acetonitrile, DMA: dimethylacetamide, DMSO: dimethyl sulfoxide, THF: tetrahydrofuran, MeOH: methanol, NMR: nuclear magnetic resonance.

interaction, which may generate the corresponding relaxed excited state (Fig. 1A. (v))[4–8,22,23]. Therefore, sensitized photoexcitation of an organic fluorophore can be performed under visible light, facilitated by stabilization of its excited state and subsequent minimization of its singlet and triplet energies at the aqueous-organic interface[23–26]. We realized that substrates like quinones (or

substrates with conjugated carbonyl groups) can participate in the dipole interaction with water to produce a stabilized excited state[27–30], and this phenomenon could be further applied in hydrogen atom transfer (HAT) to form C–C bonds[31–33]. We wanted to examine our hypothesis by exploring redox-neutral photo-Friedel-Crafts acylation and alkylation reactions of quinones. The

**Table 1 | Optimization of reaction conditions**[a, b]

| Sl. No. | Deviation from the mentioned condition | Yield (%) |
|---|---|---|
| 1. | No deviation | 80 |
| 2. | Under dark at 25 °C/80 °C | ND |
| 3. | Under air | 45 |
| 4. | Without Eosin Y | 8 |
| 5. | Only water as the solvent | 68 |
| 6. | Dry acetone as the solvent | 23 |
| 7. | Reaction continued for 18 h/36 h | 76/81 |
| 8. | Irradiation under 0.7 W 455 nm | **92** |

Reaction conditions: [a]A-1 (0.3 mmol), B-1 (0.45 mmol), EY (0.009 mmol), water/acetone (9:1) solvent mixture 3 mL (degassed), green LED (525 nm, 0.7 W), $N_2$, 24 h, 25 °C; [b]isolated yield; *ND*: not detected [by Gas Chromatography-Mass Spectrometry (GC-MS)], *LED*: Light Emitting Diode.

quinone core represents a common structural motif of a wide range of bioactive natural products (Fig. 1B). 2-Acylated-1,4-naphthohydroquinones, such as Elutherin, Isoetherin, Saintophin, or Frenolicin, have shown cytotoxicity against cancer cells[34–37]. Among various synthetic strategies, the classical Friedel-Crafts acylation of quinols is a reliable method for accessing functionalized quinone scaffolds. However, it requires oxygen-labile substrates, acyl chlorides, and stoichiometric amounts of Lewis acids. The high reactivity of the starting materials and reagents, poor regioselectivity, and the formation of unwanted side products restrict the practical application[38–40]. To avoid these shortcomings, quinones have been utilized in photo-Friedel-Crafts acylation reactions, where aldehydes are directly used as the acylating agent to produce 2-acyl quinols[41–46]. It is worth mentioning that quinones absorb in the UV region and, after excitation, can engage in single-electron transfer (SET), hydrogen atom transfer (HAT), or energy transfer (*En*T) with suitable substrates[29,30,47]. However, the mentioned photo-Friedel-Crafts reactions were performed using high-power ultraviolet (UV) light sources or direct sunlight, often requiring prolonged reaction times and complex impractical reaction equipment. In addition, the desired quinol products have a higher UV extinction coefficient than the starting quinones, they experience decomposition upon irradiation and offer inferior yields[48].

To address these shortcomings, in this report, we propose a sensitized photo-acylation and alkylation of quinones that uses visible light to initiate the reaction at the aqueous-organic interface. We observed that water's hydrogen-bonding ability at the interface causes a significant bathochromic shift in the absorption and emission spectra of quinone. Additionally, the absorption and emission spectra of Eosin Y, the dye we employed as a sensitizer, shifted hypsochromically "on water", providing a good match for photosensitization. The presented strategy resolves the recurring limitations stated in earlier reports and is generalized by demonstrating photo-Friedel-Crafts acylation and alkylation reactions with more than a hundred synthetic examples by applying an array of aromatic and aliphatic aldehydes, including several naturally occurring and drug candidates, ethers, thioethers, alkanes, silanes, and amines as the acylating or alkylating partner against quinone derivatives and maleimide.

## Results and discussions
### Spectroscopic investigations
We initiated our investigation by examining the solvatochromic behavior of 1,4-naphthoquinone (**A-1**) in various dipolar aprotic solvents, including acetonitrile (MeCN), dimethylacetamide (DMA), dimethyl sulfoxide (DMSO), tetrahydrofuran (THF), and acetone. The absorption maxima of **A-1** were observed between 320 to 332 nm in the neat solvents (Fig. 2a). Upon addition of excess polar protic solvents, such as water or methanol (MeOH), the absorption spectra shifted to longer wavelengths with a notable example where the $\lambda_{max}$ of **A-1** shifted from 330 nm to 355 nm in a water–acetone mixture (9:1). This finding was consistent for other water-organic solvent mixtures (9:1) as well. Furthermore, the emission maximum of **A-1** was also shifted from 399 nm to 428 nm when the solvent system was altered from neat acetone to a 9:1 water-acetone mixture. The bathochromic shifts can be attributed to dipolar interactions between the fluorophore (**A-1**) and solvent molecules and are explained by the Franck-Condon principle[49–51]. In polar protic solvents, the absorption transition energy decreases due to the electronic interaction field induced by the excited-state dipole, resulting in a more stabilized, relaxed photo-excited state[4,52–55]. Normalized absorption-emission plots of **A-1** reveal a bathochromic shift of the intersection from 357 nm in neat acetone to 386 nm in a water-acetone mixture (9:1), indicating a reduction in the excited-state energy of **A-1** in the aqueous environment (Fig. 2b)[56]. Similar bathochromic shifts were observed for other quinone derivatives and maleimide (see SI Fig. S3). Furthermore, to elucidate the influence of H-bonding on the water surface for **A-1**, additional [1]H and [13]C NMR experiments were conducted in different solvent mixtures. It was observed that in the [1]H NMR spectra, the alkene protons next to the carbonyl groups ($H_a$) exhibited a downfield shift from 7.05 ppm in neat acetone-$d_6$ to 6.96 ppm in a 6:1 $D_2O$/acetone-$d_6$ solvent mixture (Fig. 2e, f). Moreover, $H_a$ shifted to approximately 7.00 ppm when the solvent system was altered to either 5:2 $H_2O$/acetone-$d_6$ or 5:2 MeOH/acetone-$d_6$. A similar trend of downfield shift was observed in the case of $H_b$ as well, and it exhibited the δ values of 8.05 ppm, 7.90 ppm, 7.95 ppm and 8.02 ppm, respectively, in the solvent mixtures mentioned above (Fig. 2e). On the other hand, in the [13]C NMR spectra, the carbonyl group of **A-1** shifted from 185.7 ppm in neat acetone-$d_6$ to 178.8, 181.2, and 183.9 ppm in 6:1 $D_2O$/acetone-$d_6$, 5:2 $H_2O$/acetone-$d_6$, and 5:2 MeOH/acetone-$d_6$, respectively (Fig. 2f). These downfield shifts of the

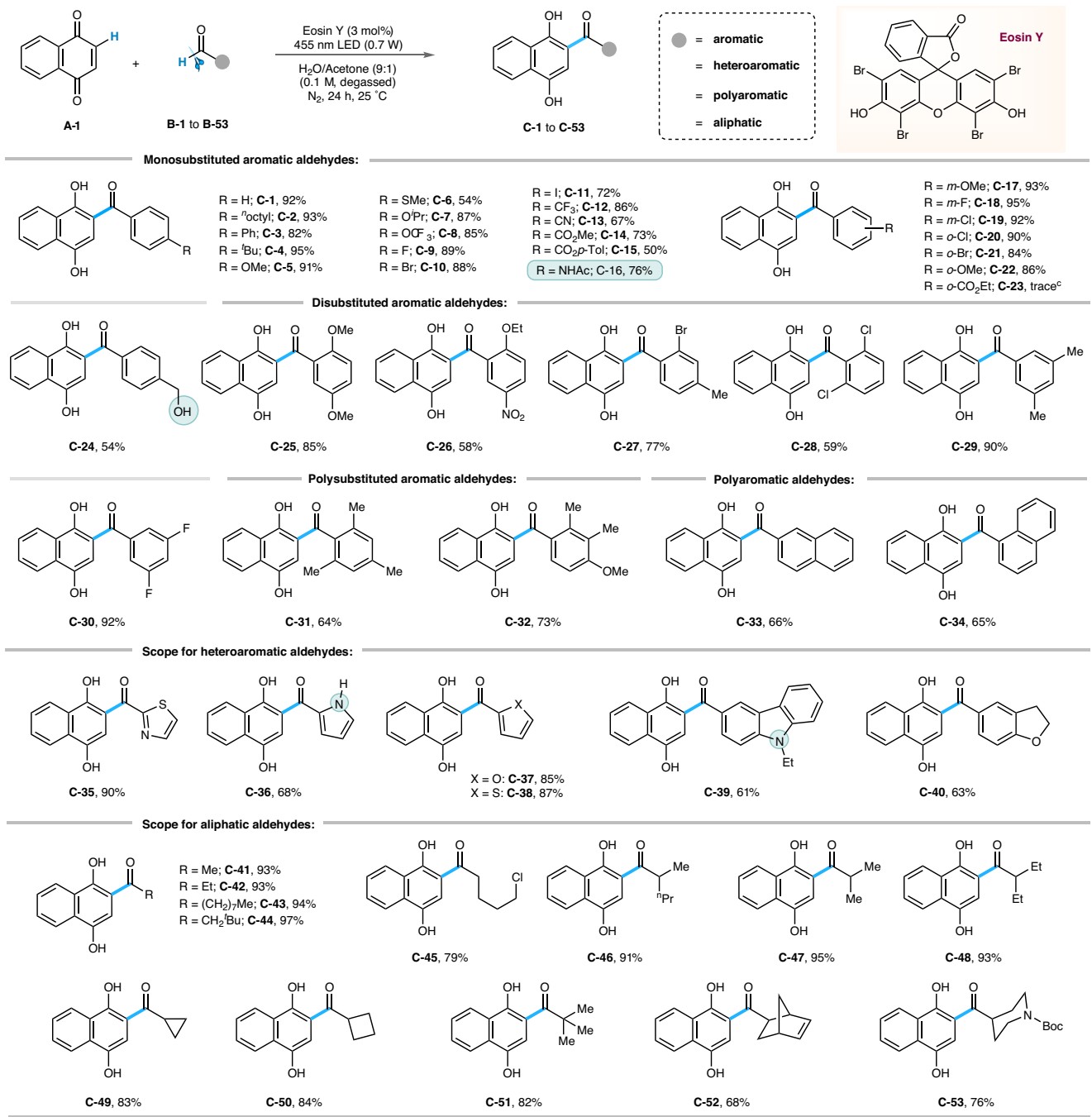

**Fig. 3 | Scope of aromatic, heteroaromatic, and aliphatic aldehydes:[a,b].** Reaction conditions: [a]**A-1** (0.3 mmol), **B** (0.45 mmol), Eosin Y (0.009 mmol), water/acetone (9:1) solvent mixture 3 mL (degassed), blue LED (455 nm, 0.7 W), N$_2$, 24 h, 25 °C. [b]Isolated yield. [c]Detected by HRMS (High Resolution Mass Spectroscopy).

olefinic and aromatic protons (*ortho* to the carbonyl group of **A-1**) and carbonyl carbons are consistent with hydrogen-bonding interactions involving the naphthoquinone moiety, which further supports our working hypothesis[17]. To achieve visible-light sensitization of quinone **A-1** in the aqueous reaction medium, we selected Eosin Y (EY) as a photosensitizer due to its large excitation window and water solubility. A hypsochromic shift in the absorption maxima and a significant increase in the extinction coefficient were observed in water-acetone (9:1) compared to the neat organic solvent. The increase in the extinction coefficient is explained in the literature by a possible aggregation of Eosin Y in the aqueous environment, while the negative

solvatochromism implies destabilization in the excited state compared to the ground state, likely due to local (e.g., hydrogen bonding) and global polarity effects. The fluorescence emission band of EY exhibited a blue shift when the solvent was changed from pure acetone to a water-acetone (9:1) mixture[57,58]. Accordingly, a normalized absorption-emission plot for Eosin Y shows a noticeable hypsochromic shift of the intersection point from 548 nm in neat acetone to 538 nm in a water-acetone mixture (9:1), which suggests an increase in the excited-state energy of Eosin Y in the aqueous environment (Fig. 2c). In addition to that, the experimentally observed bathochromic shift of absorption and emission of **A-1** in the presence of water was further confirmed by

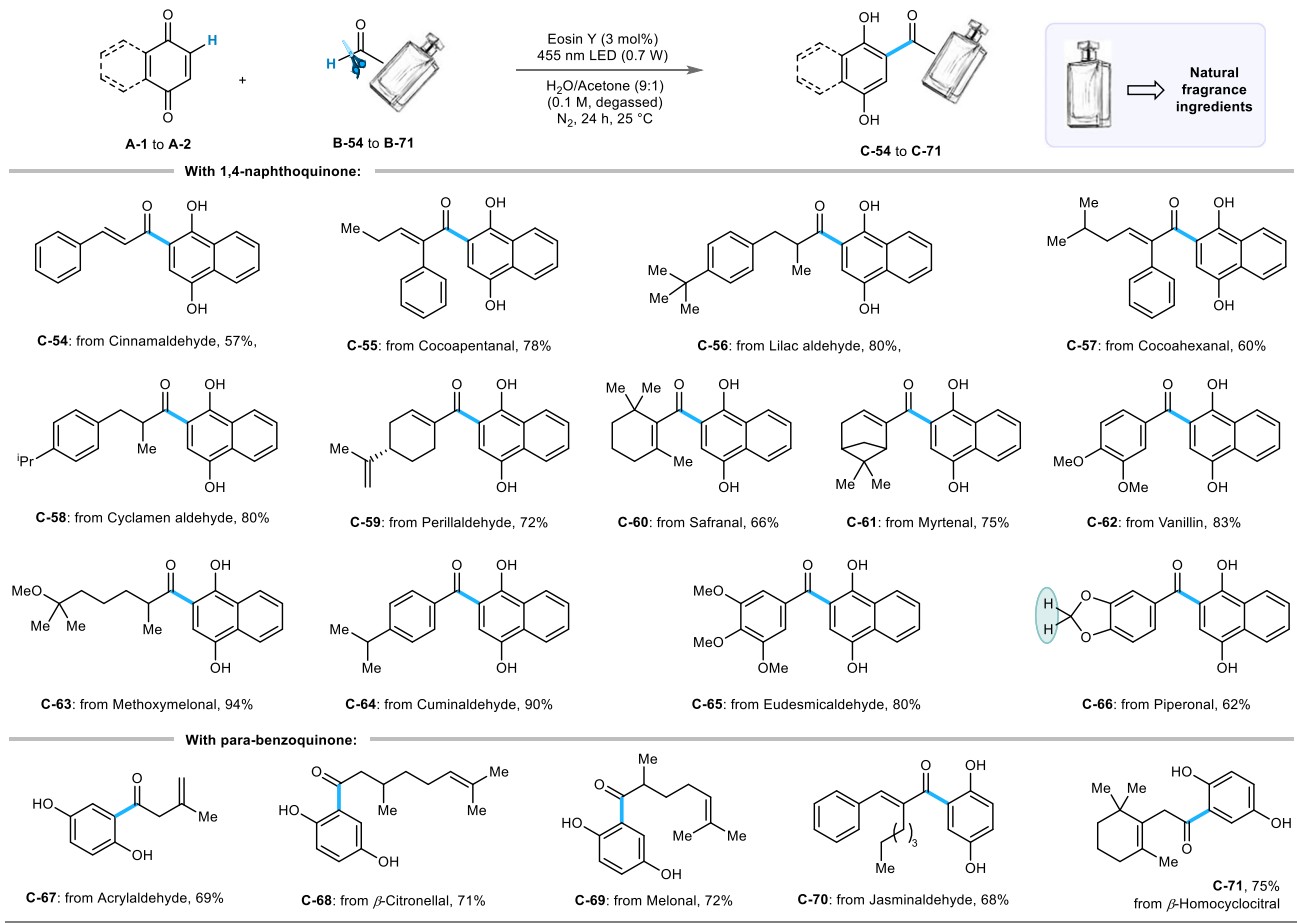

**Fig. 4 | Scope of naturally occurring aldehydes and fragrance ingredients:[a, b].** Reaction conditions: [a]**A-1** or **A-2** (0.3 mmol), **B** (0.45 mmol), EY (0.009 mmol), water/acetone (9:1) solvent mixture 3 mL (degassed), blue LED (455 nm, 0.7 W), $N_2$, 24 h, 25 °C. [b]Isolated yield.

our computational studies, which suggest the stabilization of the photo-excited states (both the singlet and triplet) for **A-1** under an aqueous environment (see SI Fig. S7 to S9 for detailed calculations). However, the effect of water on the electronic structure of Eosin Y is more complex, as it may affect tautomeric equilibria and their individual electronics, which go beyond the scope and the prime interest of this work[59−62]. The complementary bathochromic absorption shift of quinones and hypsochromic shift of the sensitizer Eosin Y at the aqueous-organic interface make them a perfect match to facilitate the sensitized HAT process under visible light. A luminescence quenching study of Eosin Y by **A-1** was conducted in dry acetone and a water-acetone mixture (9:1). Stern-Volmer analysis (Fig. 2d) revealed pronounced quenching of the photosensitizer by **A-1** in a water-acetone (9:1) solvent mixture.

### Reaction development and optimization studies

To apply this interesting spectroscopic outcome, we explored the Friedel-Crafts (F-C) acylation by treating **A-1** and **B-1** in the presence of Eosin Y (EY) as the photosensitizer, using a water-acetone solvent mixture (9:1) under green LED (0.7 W, 525 nm) at 25 °C. To our delight, after 24 hours of irradiation under an inert atmosphere, **C-1** was obtained in 80% isolated yield (Table 1, entry 1) [Please see the detailed optimization study in SI Tables 1 to 4]. The reaction ceased in the absence of light, even at elevated temperature (entry 2). The yield dropped when the reaction was performed under air (entry 3, 45%) or

without Eosin Y (entry 4, 8%). The formation of **C-1** decreased to 23% (entry 6) when the reaction was carried out in dry acetone, without changing any other parameters. Extending the reaction time to 18 h or 36 h had little effect on the reaction outcome (entry 7). Irradiation of the reaction mixture under 455 nm (0.7 W) improved the isolated yield of **C-1** to 92%.

### Evaluation of the scope for Friedel-Crafts acylation

After establishing the optimized reaction conditions, we explored the scope of the acylation reaction (Fig. 3). Mono-substituted aromatic aldehydes bearing electron neutral phenyl or alkyl groups (**C-1** to **C-4**), electron rich -OMe, -SMe, -O[i]Pr, -OCF₃ groups (**C-5** to **C-8, C-17**, and **C-22**), all types of halogens (**C-9** to **C-11** and **C-18** to **C-21**) and electron withdrawing -CF₃, -CN, -COOR (R = Me, Et, *p*-Tol) groups (**C-12** to **C-15** and **C-23**) at the *para-*, *meta-*, and *ortho-* positions were converted to the corresponding 2-acyl quinols in good to excellent yields (50% to 95%). However, a trace amount of expected product was formed in the case of **C-23**, which is probably due to a different steric and electronic factor that operates for 2-substituted aryl aldehydes bearing an electron-withdrawing group. Notably, aromatic rings containing a protected amine (-NHAc) and a free hydroxyl group were retained during the transformation and gave **C-16** and **C-24** in 76% and 54% yield, respectively. Di- and tri-substituted aromatic aldehydes also afforded the desired conversion to **C-25** to **C-32** in 58% to 73% isolated yields. The nitro group was also tolerated, and **C-26** was obtained in

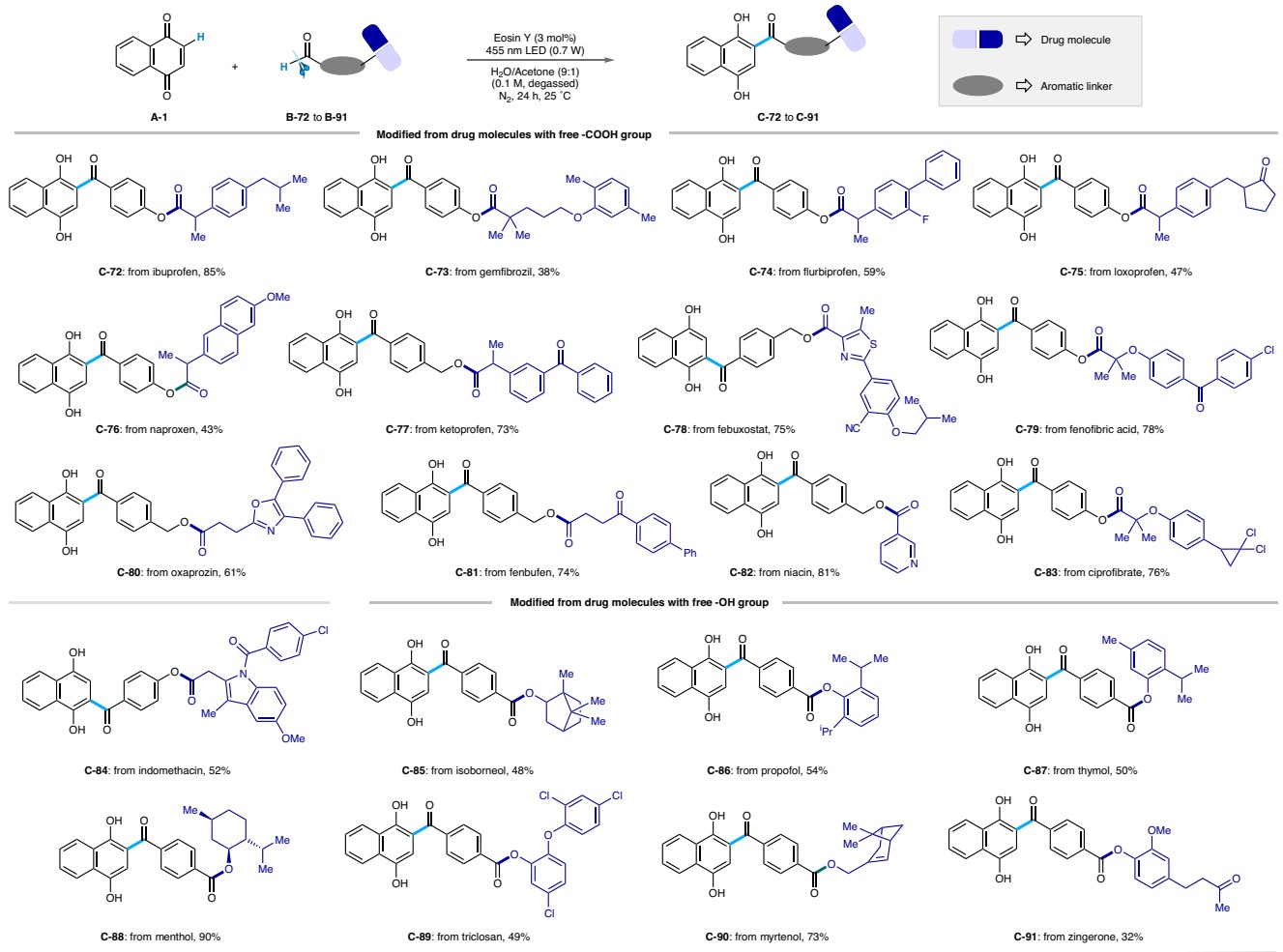

**Fig. 5 | Late-stage modifications with drug molecules:[a, b].** Reaction conditions: [a]**A-1** (0.3 mmol), **B** (0.45 mmol), EY (0.009 mmol), water/acetone (9:1) solvent mixture 3 mL (degassed), blue LED (455 nm, 0.7 W), N₂, 24 h, 25 °C. [b]Isolated yield.

58% yield. The standard reaction conditions were applied to polyaromatic and heteroaromatic aldehydes (including *NH*-pyrrole), and the corresponding hydroacylated quinols (**C-33** to **C-40**) were obtained in moderate to good yields (63% to 90%). Pleasingly, the reaction outcome was also uniform across a range of structurally diverse aliphatic aldehydes, including strained cyclopropane (**C-49**) and cyclobutene (**C-50**) scaffolds, and the transformations occurred smoothly to produce **C-41** to **C-53** in excellent yields of up to 97%. We further extended the scope of hydroacylation reactions of 1,4-naphthoquinone (**A-1**) and *p*-benzoquinone (**A-2**) by exploiting an array of naturally occurring aliphatic and aromatic aldehydes, which are important in the food and cosmetic industries (Fig. 4). In this regard, natural fragrance materials, such as cinnamaldehyde (**C-54**), cocoapentanal (**C-55**), lilac aldehyde (**C-56**), cocoahexanal (**C-57**), cyclamen aldehyde (**C-58**), perillaldehyde (**C-59**), safranal (**C-60**), myrtenal (**C-61**), vanillin (**C-62**), methoxymelonal (**C-63**), cuminaldehyde (**C-64**), eudesmicaldehyde (**C-65**), piperonal (**C-66**), acrylaldehyde (**C-67**), β-citronellal (**C-68**), melonal (**C-69**), jasminaldehyde (**C-70**), β-homocyclocitral (**C-71**) were converted to the desired products in commendable yields (57% to 94%).

**Late-stage modifications**
Next, we applied our Friedel-Crafts acylation strategy to a range of pharmaceutically relevant bioactive compounds (Fig. 5). Drugs containing the synthetic handle, such as a carboxylic acid or a hydroxyl group, were first attached to compatible benzaldehyde derivatives via traditional esterification reactions. The resultant aryl aldehydes were then subjected to optimized reaction conditions using **A-1**. Synthetically modified drug candidates and natural products such as ibuprofen (**C-72**), gemfibrozil (**C-73**), flurbiprofen (**C-74**), loxoprofen (**C-75**), naproxen (**C-76**), ketoprofen (**C-77**), febuxostat (**C-78**), fenofibric acid (**C-79**), oxaprozin (**C-80**), fenbufen (**C-81**), niacin (**C-82**), ciprofibrate (**C-83**), indomethacin (**C-84**), isoborneol (**C-85**), propofol (**C-86**), thymol (**C-87**), menthol (**C-88**), triclosan (**C-89**), myrtenol (**C-90**), and zingerone (**C-91**) were converted in 38% to 90% yield to the corresponding hydroacylated products.

**Scope of quinone and other conjugated carbonyl moieties**
We then examined the applicability of the method to various quinone scaffolds (Fig. 6). The hydroacylation reaction was successfully carried out when *para*-benzoquinone (**A-2**) was reacted with electronically distinct aromatic and aliphatic aldehydes, delivering **C-92** to **C-98** in decent yields (65% to 80%).

Notably, an inseparable regio-isomeric mixture (**C-99** and **C-99′**) was found in 76% of isolated yield in case of 2-methyl benzoquinone (**A-3**). Surprisingly, 2-acylquinones (**C-100**, 67% and **C-101**, 65%) were obtained as the sole products when 2-chloro (**A-4**) and 2-bromonaphthoquinone (**A-5**) were reacted under standard conditions with benzaldehyde and 4-chlorobenzaldehyde. Naturally

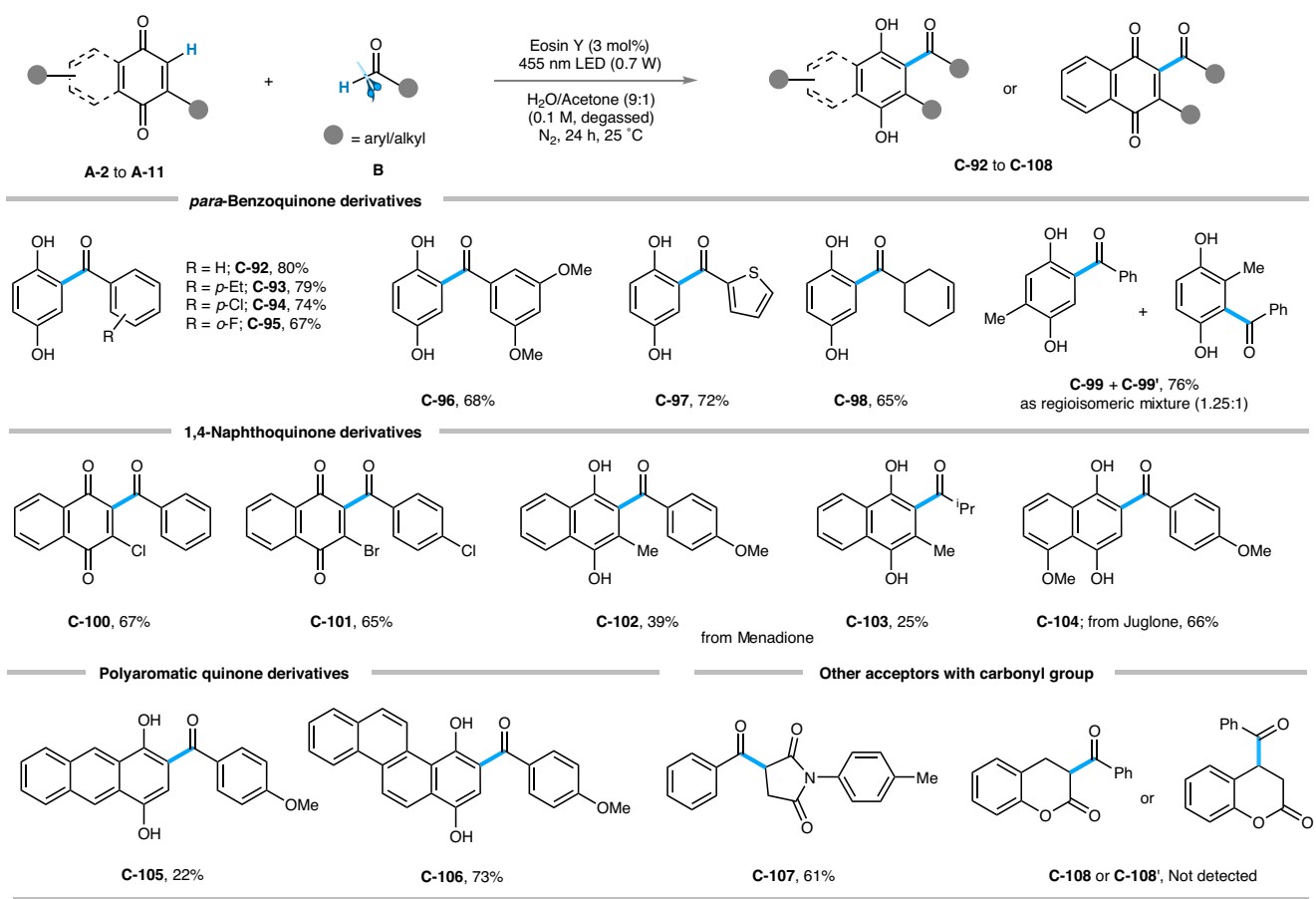

**Fig. 6 | Scope of quinone and di-carbonyl derivatives:**[a,b]. Reaction conditions: [a]**A** (0.3 mmol), **B** (0.45 mmol), EY (0.009 mmol), water/acetone (9:1) solvent mixture 3 mL (degassed), blue LED (455 nm, 0.7 W), N$_2$, 24 h, 25 °C. [b]Isolated yield. ND: Not detected (by GC-MS).

occurring quinones like menadione (**A-6**) and juglone (**A-7**) also reacted in the hydroacylation with both aromatic and aliphatic aldehydes to accomplish **C-102** to **C-104** in 25% to 66% yields. Polyaromatic quinone derivatives like anthracene-1,4-dione (**A-8**) and chrysene-1,4-dione (**A-9**) displayed a similar reactivity pattern to offer **C-105** and **C-106**, respectively, in 22% and 73% isolated yields. Apart from quinones, we were interested in investigating other acceptors, such as *N*-aryl maleimide (**A-10**), which gave **C-107** in 61% isolated yield. However, coumarin did not participate in the acylation reaction under the standard reaction conditions (remained unreacted), and neither of the anticipated products (**C-108** or **C-108'**) was detected by GC-MS (Gas Chromatography-Mass Spectrometry).

### Evaluation of the scope for Friedel-Crafts alkylation
Next, we attempted photo-Friedel-Crafts alkylation reactions of quinones, which was elusive in earlier attempts (Fig. 7). We reacted **A-1** with a library of coupling partners that possess comparatively reactive C–H bonds (BDE ~ 84 to 96 kcal/mol) and tend to undergo C–H HAT to generate a carbon-centered radical. Satisfyingly, under the developed conditions, different ethers and thioethers participated in the hydroalkylation reaction yielding the desired products for tetrahydrofuran (**C-109**), phthalan (**C-110**), tetrahydropyran (**C-111**), isochroman (**C-112**), 1,4-dioxane (**C-113**), 1,3,5-trioxane (**C-114**), benzo[*d*][1,3]dioxole (**C-115**), protected benzyl alcohol (**C-116**), isopropoxybenzene (**C-117**), tetrahydrothiophene (**C-118**), and isopropyl(phenyl)sulfane (**C-119**) in 50% to 80% of isolated yields. Aryl and alkyl silanes show C–Si bond forming[63] reactions with **A-1** to furnish **C-120** to **C-122** descent yields. The scope for Friedel-Crafts alkylation was further extended using partially saturated

carbocycles, which include anthrone, fluorene, indan-1-one, celestolide (fragrance material), and phenylcyclopropane, which were transformed to the corresponding 2-alkyl quinols (**C-124** to **C-128**) in moderate to good yields (56% to 85%). Additionally, acyclic aromatic hydrocarbon moieties such as diphenylmethane, cumylbenzene, *sec*-butyl benzene also delivered the expected products **C-129** to **C-133** in 53% to 92% isolated yields. However, the alkylation reaction was not successful for toluene derivative (**C-123**), as it was oxidized to the corresponding aldehyde and acylation occurred. Furthermore, the photosensitized HAT process transformed cyclic amines into **C-134** and **C-135** via the formation of α-amino radical intermediates (which may arise from photo-oxidation and proton elimination), affording yields of 66% and 54%, respectively. However, inferior productivity was observed for *N*-phenyl tetrahydropyrrole, and only a trace amount of **C-136** was detected by GC-MS. This is probably due to the photochemical oxidation of the electron-rich amine center of *N*-phenyltetrahydropyrrole, where SET is more likely facilitating than HAT to generate an iminium cation under the standard reaction conditions.The corresponding iminium cation underwent a nucleophilic water attack and a successive oxidation to release *N*-phenylpyrrolidone (detected by GC-MS)[64–66].

### Synthetic utilities
We performed several syntheses on a larger scale (Fig. 8A) under standard reaction conditions (7 mmol, 1.1 g). Pleasingly, the photocatalyzed hydroacylation on a gram scale yielded products **C-1**, **C-41**, and **C-137** in 77%, 60%, and 72% isolated yields, respectively. Some of the products were further derivatized to illustrate applications.

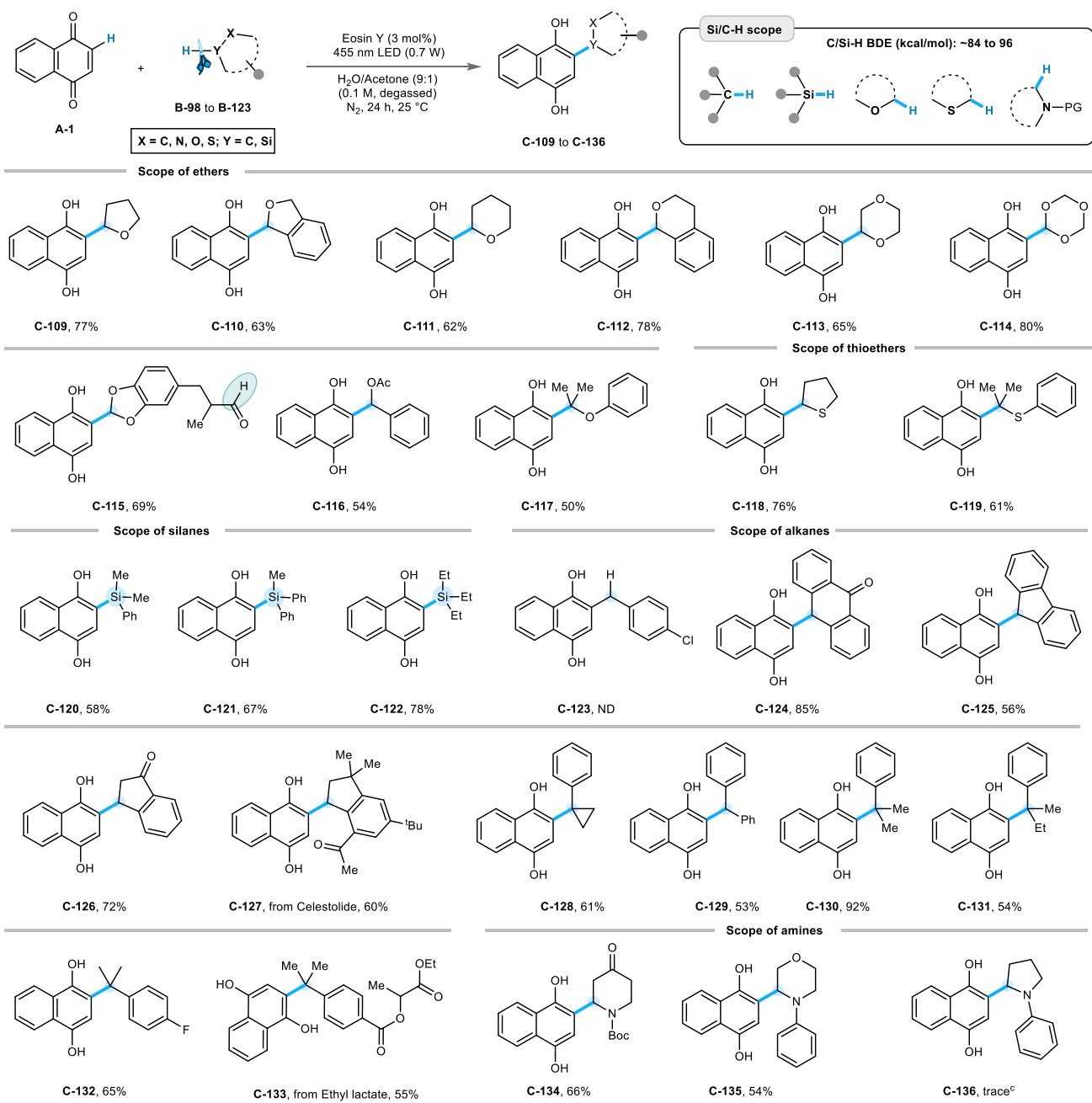

**Fig. 7 | Scope of Friedel-Crafts alkylation:**[a, b]. Reaction conditions: [a]**A-1** (0.3 mmol), **B** (0.9 mmol), EY (0.009 mmol), water/acetone (9:1) solvent mixture 3 mL (degassed), blue LED (455 nm, 0.7 W), N$_2$, 24 h, 25 °C. [b]Isolated yield. [c]Detected in GC-MS.

Accordingly, **C-1** was subjected to oxidation of the quinol structural motif using Ag$_2$O to afford 2-benzoyl quinone (**D-1**), exclusively (Fig. 8B (i))[67]. In contrast, reduction of **C-1** using NaBH$_4$ offered **D-2** in 90% yield (Fig. 8B (ii))[68]. Next, compound **C-1** was reacted with phenylacetyl chloride under basic conditions to give a substituted angular benzocoumarin scaffold (**D-3**) in 68% yield (Fig. 8B (iii))[68]. Interestingly, fluorinated product, **C-137**, underwent an intramolecular aromatic nucleophilic substitution reaction (S$_N$Ar) after treatment under basic conditions to yield a substituted angular xanthone derivative (**D-4**) in 84% yield (Fig. 8B (iv))[68]. Lastly, we successfully constructed the key intermediate (**D-5b**) for the natural product *Balsaminone A* by reacting **C-41** sequentially with 2,3-dichloro-1,4-naphthoquinone followed by benzyl bromide (Fig. 8B (v))[69].

## Mechanistic studies and proposed reaction mechanism

To elucidate the reaction mechanism, we performed several control experiments (Fig. 9). Eosin Y photosensitization was confirmed by inhibiting the model reaction (Fig. 9A) in the presence of well-known triplet quenchers, such as trans-stilbene ($E_T$ = 49.3 kcal/mol) or azobenzene ($E_T$ = 35.4 kcal/mol)[70,71]. Under certain reaction conditions, Eosin Y may also act as a HAT agent[72]. To exclude this reaction pathway, the standard reaction was performed with other photosensitizers, such as [Ir] or [Ru] photocatalysts, which are known to exhibit sufficiently high triplet energies, yielding comparable yields of product **C-1** (Fig. 9B)[70,71]. Next, we performed standard reactions in the presence of radical scavengers, such as TEMPO and 1,1-diphenyl ethylene, which significantly reduced the product yield.

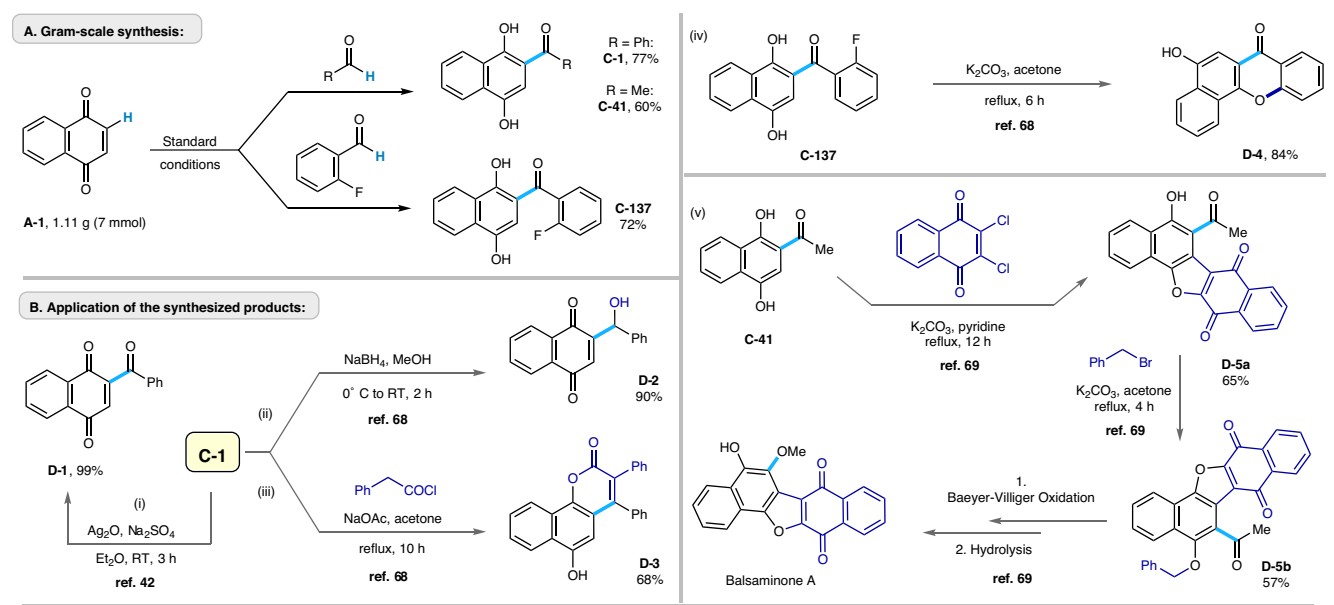

**Fig. 8 | Application of the synthetic protocol. A** scale-up synthesis for 2-acyl hydroquinone derivatives; **B** derivatization of the synthesized products for **C-1, C-137,** and **C-41.**

Intermediates, such as **E-1** and **E-2**, were detected by HRMS. These experiments suggest the presence of an acyl radical intermediate in the photo-Friedel-Crafts acylation process (Fig. 9C). To determine the rate-determining step of the overall transformation, two parallel reactions were run under standard reaction conditions using normal benzaldehyde (**B-1**) and benzaldehyde-$d_6$ (**B-1'**). After 12 hours, the crude reaction mixture was analyzed by ¹H NMR using $CH_2Br_2$ as the internal standard. 63% of **C-1** was formed, whereas **C-1'** was produced only in 24% at 12 hours of reaction time, and a KIE of 2.63 was calculated (Fig. 9D). This experiment indicates that hydrogen atom abstraction from the C–H bond of the coupling partner may be the rate-determining step of the overall reaction (See SI Fig. 2 for kinetic plots).

Finally, based on the spectroscopic findings and control experiments, we have proposed a plausible reaction mechanism for the Friedel-Crafts reaction of quinones (Fig. 9E). Upon photosensitization in the presence of Eosin Y, water H-bonded quinone is converted to its triplet state and abstracts a H-atom from the corresponding acyl or alkyl coupling partner (**B**) to produce the radical intermediate **2A** and carbon-centered radical **1B**. The hydroquinone radical, via its resonance structure **3A**, combines with the acyl or alkyl radical **1B** to give **4A**. Lastly, a keto-enol tautomerism facilitates the aromatization to yield the hydro-functionalized quinols **C**.

In summary, we employed the concept of excited-state stabilization of a chromophore via H-bonding and dipole interaction "on-water" for the redox-neutral Friedel-Crafts acylation and alkylation of quinones. Comprehensive spectroscopic studies provided evidence of H-bonding at the aqueous-organic interface, which enables a simultaneous bathochromic shift for quinones (by excited-state stabilization) and a hypsochromic shift of the photosensitizer, Eosin Y, making them an appropriate pair for visible-light photosensitization. The practical viability of the photo-Friedel-Crafts reactions was demonstrated with aromatic and aliphatic aldehydes, ethers, thioethers, alkanes, silanes, and amines, including some naturally occurring substances and diversifications of biomolecules, as well as reactions on a gram scale. The proposed mechanism is experimentally supported by spectroscopic observations, computational studies, and control experiments. The aqueous-organic interface offers advantages for photo-Friedel-Crafts reactions that can be performed under visible light, thereby overcoming the practical limitations of product stability and versatility, as previously reported.

## Methods

### "On-water" Eosin Y catalyzed redox-neutral Friedel-Crafts acylation of quinones

A 5 mL crimp top vial was charged with 1,4-quinone derivative (**A**, 0.30 mmol, 1.0 equiv), aromatic or aliphatic aldehyde (**B**, 0.45 mmol, 1.5 equiv), and Eosin Y (6 mg, 0.009 mmol, 0.03 equiv). Then the vial was crimped. The vial was degassed and refilled with nitrogen using the Schlenk line technique (3 times). A pre-degassed (15 min) mixture of water/acetone (9:1, 3 mL) was added to the reaction vial and stirred at room temperature for 24 h under the irradiation of a single blue LED ($\lambda_{max} = 455 \pm 15$ nm). Upon completion, the reaction mixture was diluted with EtOAc (20 mL), washed with saturated $Na_2CO_3$ solution (15 mL, twice), followed by brine (15 mL, once), and extracted with EtOAc. The combined organic layer was dried over $Na_2SO_4$, concentrated under vacuum, and the residue was purified by silica gel (100–200 mesh) column chromatography using a mixture of PE/EtOAc as the eluent, yielding pure products.

### "On-water" Eosin Y catalyzed redox-neutral Friedel-Crafts alkylation of quinones

A 5 mL crimp top vial was charged with 1,4-quinone derivative (**A**, 0.30 mmol, 1.0 equiv), alkyl coupling partner (**B-98** to **B-123**), 0.9 mmol, 3.0 equiv), and Eosin Y (6 mg, 0.009 mmol, 0.03 equiv). Then the vial was crimped. The vial was degassed and refilled with nitrogen using the Schlenk line technique (3 times). A pre-degassed (15 min) mixture of water/acetone (9:1, 3 mL) was added to the reaction vial and stirred at room temperature for 24 h under the irradiation of a single blue LED ($\lambda_{max} = 455 \pm 15$ nm). Upon completion, the reaction mixture was diluted with EtOAc (20 mL), washed with saturated $Na_2CO_3$ solution (15 mL, twice), followed by brine (15 mL, once), and extracted with EtOAc. The combined organic layer was dried over $Na_2SO_4$, concentrated under vacuum, and the residue was purified by silica gel (100-200 mesh) column chromatography using a mixture of PE/EtOAc as the eluent, yielding pure products.

Full experimental procedures are provided in the supplemental information.

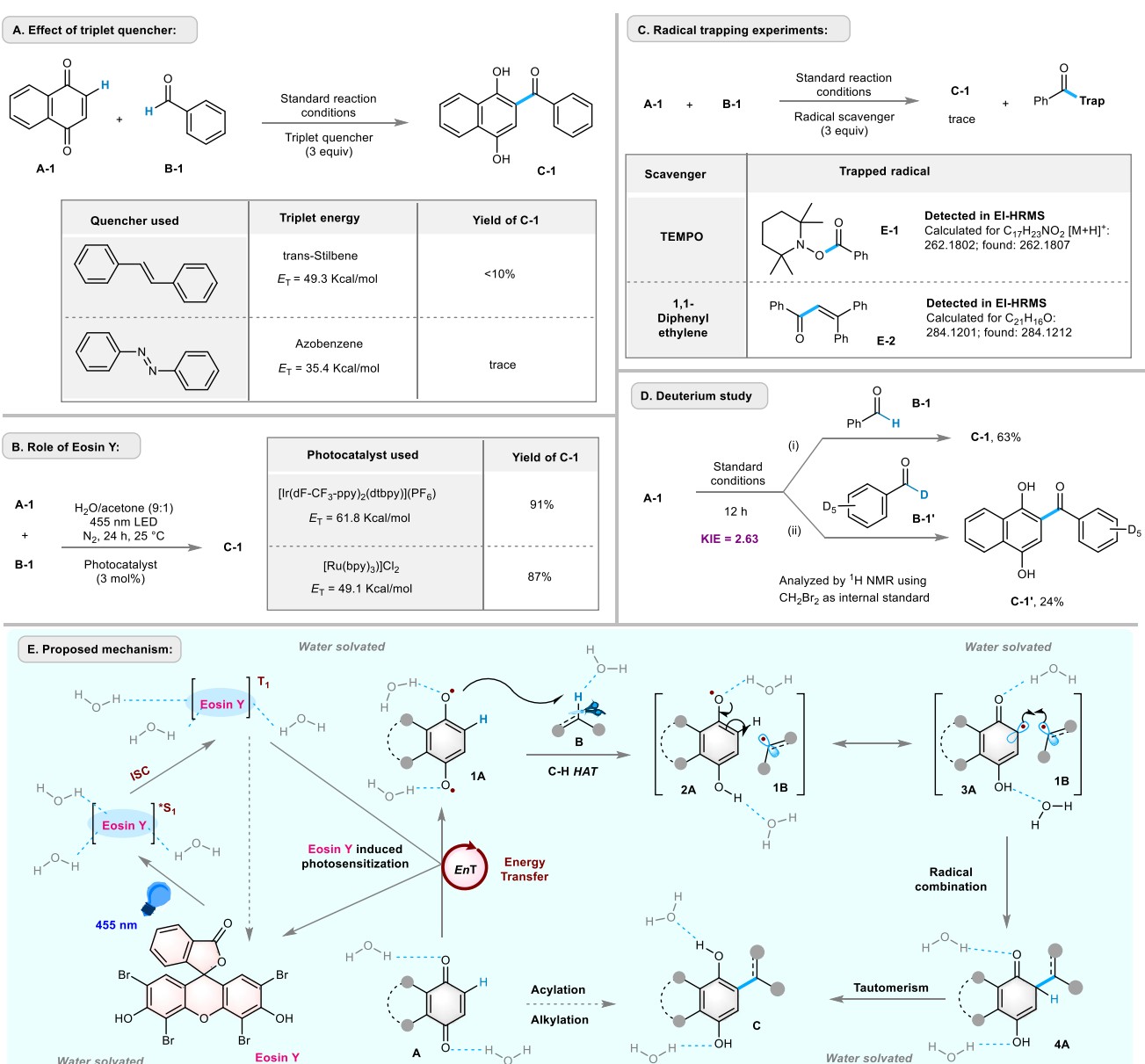

**Fig. 9 | Mechanistic studies and proposed mechanism. A** reaction outcome in presence of triplet-energy quencher; **B** elucidating the photocatalytic role of Eosin Y; **C** trapping the radical intermediate(s) involved in the reaction; **D** kinetic study to check the rate-determining step (RDS); **E** plausible mechanism for Photo-Friedel-Crafts alkylation and acylation; $E_T$: triplet energy, *En*T: energy transfer, *EI-HRMS*: electron ionization high-resolution mass spectrometry, NMR: nuclear magnetic resonance, TEMPO: 2,2,6,6-Tetramethylpiperidin-1-yl)oxyl free radical, KIE: kinetic isotope effect, HAT: hydrogen atom transfer.

## Data availability
The NMR and HRMS data generated in this study have been deposited in Rader4chem https://radar4chem.radar-service.eu/radar/de/dataset/z1v8szzsfn45kdw6. The processed NMR and other spectroscopic data generated in this study are provided in the Supplementary Information. Data supporting the findings of this manuscript are also available from the corresponding author upon request.

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

## Acknowledgements

This work was supported by the Deutsche Forschungsgemeinschaft [DFG (German Science Foundation) grants TRR 325-444632635 and KO 1537/27-1]. The authors thank Mrs. Julia Zach (University of Regensburg) and Dr. Rudolf Vasold (University of Regensburg) for their help with fluorescence and GC-MS measurements, respectively, Mrs. Dana Králová for assistance with NMR measurements, Mr. Hriday Kumar Pal for helping in the graphics design, and Ernst Lautenschlager for his help in resolving technical issues. We also thank the central NMR and HRMS facility at the University of Regensburg for their support in NMR and HRMS measurements.

## Author contributions

T.M., R.S., and B.K. conceived the project, and B.K. supervised the project. T.M. and R.S. designed the experiments, with input from B.K. T.M. and R.S. performed all the optimization, scope, and mechanistic study, and analyzed the results. T.M., R.S., and B.K. wrote the manuscript with input from all the authors. E.M.V. and J.R. performed the computational studies. T.M. and R.S. contributed equally to this work.

## Funding

## Competing interests

The authors declare no competing interests.
