## [Transparent Peer Review file · Nature Communications]

“On-Water” Photosensitization Enables Redox Neutral Acylation and Alkylation of Quinones

Corresponding Author: Professor Burkhard König

Version 0:

Reviewer comments:

Reviewer #1

(Remarks to the Author)

König et al. disclose a redox-neutral photo-Friedel–Craft acylation/alkylation of quinones that proceeds at the aqueous–organic interface through H-bonding and dipole interactions. The protocol tolerates a broad substrate scope and has been validated on gram scale; downstream derivatizations underscore its synthetic utility. Spectroscopic studies were provided to rationalize the enhancement of reactivity, yet the mechanistic rationale remains incomplete.

Overall, I support the publication of this work in after minor revisions.

1. Apart from spectral analysis, more evidences are suggested to be provided to support the proposed H-bonding and dipole interaction, such as NMR.
2. For the coumarin acylation, the product depicted is the α -acylated isomer (C108). Please clarify why the β -acylated regioisomer—expected on the basis of the mechanistic argument—is not observed.
3. N-Phenyltetrahydropyrrole affords only trace amounts of C-136. Discuss whether this outcome arises from steric shielding, diminished H-bonding ability, or competing side reactions.
4. Diaryl ketones are common triplet photosensitizers. When aryl aldehydes are used, the corresponding ketone products could conceivably act as internal sensitizers. Report Stern–Volmer quenching data, triplet-energy calculations, or sensitizer-free control reactions to exclude product-sensitized pathways.

Reviewer #2

(Remarks to the Author)

The submitted manuscript by König and co-workers presents another application of on-water photochemistry in organic synthesis and follows up the original paper from the group in Science. The authors present the concept of influencing excited states by the water environment based on hydrogen bonds. This effect is manifested both on the substrate (benzoquinone and other quinones) and on the catalyst (EOSIN Y), and allows the activation of the substrate by energy transfer. Subsequently, acylation/alkylation of the substrate can occur, which is based on the abstraction of hydrogen from the aldehyde or precursor of stable radicals. The authors document the effect of water on the absorption/fluorescence properties. Subsequently, they use an optimized methodology to prepare a series of acylated/alkylated quinones with various aldehydes or alkylating agents. Small-scale experiments are supplemented by experiments on a gram scale. The reactions tolerate various substrates and functional groups, with the exception of alkylating agents producing unstable radicals. Overall, I conclude that the work is clearly and comprehensibly written, most (see below for other comments) conclusions are clearly supported by measured data. Since this is a new and rapidly developing topic and the article, even in its current state, provides a guarantee (after minor edits) of an important message of interest to a wide range of readers, I recommend accepting the article for publication in Nature Communication after minor revision.

Other comments:

1/ Authors claim (see abstract) that water surface reduces triplet state energy of quinone substrate and increases triplet state energy of Eosin Y. On my own, it can be hardly concluded from data reported. Authors measured effect of water on absorption and fluorescence emission spectra but it can show only effect on singlet excited state. Of course, also triplet state energy might be changed but there is no direct evidence.

2/ Regarding substrate scope of alkylation (Table 6), ethylbenzene derivative is missing. Only toluene and then trisubstituted derivatives were proven. It would be interesting to see whether switch to secondary derivative giving secondary radical is enough to provide the reaction.

3/ Blue point is missing in legend in Figure 2D. Note, description A...D is not shown.

4/ Only traces of product are given in Table 2 for C23. But it is not commented accordingly in the text.

5/ Naphthaquinone should be naphthoquinone on page 7 down.

6/ Figure 3a is missing in the main text.

7/ I suggest catalytic role of EY should be better shown in Figure 3. By excitation and ISC, triplet excited state is formed which likely provides EnT.

Reviewer #3

(Remarks to the Author)

This manuscript describes a photo-induced, redox-neutral Friedel-Craft acylation and alkylation of quinones under water-oil interfaces in the presence of Eosin Y as photosensitizer. The method operates under transition-metal-free conditions, shows high substrate adaptability, and gives access to functionalized quinone products of potential synthetic value. The reaction design is conceptually appealing, and the operational simplicity makes it attractive. I recommend acceptance of this work for publication, upon addressing the following minor issues:

1. In Table 2, the reason for lower yields for substrates like C-23, C-53 was not discussed. Did the author observe byproducts?

2. The authors should note that C-134 and C-135/C-136 are not the same class of compounds. I support the statement that C-134 can generate an α -amino radical via light-mediated HAT. However, applying the same description to C-135 and C-136 — that they also generate an α -amino radical via light-mediated HAT — may not be appropriate. For C-135 and C-136, a single-electron transfer (SET) pathway is likely more favorable than HAT.

In addition, for C-136, no obvious product was observed. What might be the possible reasons for this outcome? Is it necessary to further examine the reactivity of the N-phenylpiperidine substrate? For this type of substrate, the authors may need to discuss the mechanism of α -amino radical generation separately.

3. The scope of the reaction may be further illuminated by two key references (Angew. Chem. Int. Ed. 2016, 55, 4040; Chem. Sci. 2016, 7, 7002).

4. Please verify that all citations and labels in Scheme 1 correspond accurately to their sources in the text and reference list.

Version 1:

Reviewer comments:

Reviewer #1

(Remarks to the Author)

The authors have addressed all the issues raised. The revised manuscript is recommended for acceptance.

Reviewer #2

(Remarks to the Author)

The manuscript represents a significant contribution to a highly appealing topic in the field of photoredox catalysis. During revision, the authors carefully corrected and supplemented the original manuscript according to the comments of the reviewers. I can now happily recommend the manuscript as suitable for publication in Nature Communication.

Reviewer #3

(Remarks to the Author)

The author has addressed most of my concerns, and the manuscript can be published in its current form in Nature Communications.

Our manuscript was reviewed by three expert referees, and we are pleased to receive their positive comments. Based on their feedback, we have revised the manuscript to address the reviewers' concerns. We conducted a new set of experiments and accordingly modified the SI. We found the reviewers' comments very helpful, and they assisted us in improving our manuscript. A point-by-point response to the reviewers' comments is enclosed. Additionally, we have uploaded a separate revised manuscript file with the major changes highlighted in yellow to track the modifications from the original version.

Point by point reply:

Reviewer #1 (Remarks to the Author):

König et al. disclose a redox-neutral photo-Friedel–Craft acylation/alkylation of quinones that proceeds at the aqueous–organic interface through H-bonding and dipole interactions. The protocol tolerates a broad substrate scope and has been validated on gram scale; downstream derivatizations underscore its synthetic utility. Spectroscopic studies were provided to rationalize the enhancement of reactivity, yet the mechanistic rationale remains incomplete.

Overall, I support the publication of this work in after minor revisions.

Comment:

1. Apart from spectral analysis, more evidences are suggested to be provided to support the proposed H-bonding and dipole interaction, such as NMR.

Our response: We thank the reviewer for this suggestion. We performed ^1H and ^{13}C NMR studies using 1,4-naphthoquinone (A-1) individually in neat acetone- d_6 , 6:1 mixture of D_2O / acetone- d_6 , 5:2 mixture of H_2O / acetone- d_6 and 5:2 mixture of MeOH/ acetone- d_6 , respectively, to support the H-bonding interaction of quinone in water/acetone mixture. The following observation is now added to the revised manuscript (Figure 2e and 2f).

“Furthermore, to elucidate the influence of H-bonding on water surface for A-1, additional ^1H and ^{13}C NMR experiments were conducted in different solvent mixture. It was observed that in the ^1H NMR spectra, the alkene protons next to the carbonyl groups (H_a) exhibited a downfield shift from 7.05 ppm in neat acetone- d_6 to 6.96 ppm in a 6:1 D_2O /acetone- d_6 solvent mixture (See NMR spectra in SI for details). Moreover, H_a shifted approximately to 7.00 ppm when the solvent system was altered to either 5:2 H_2O /acetone- d_6 or 5:2 MeOH/acetone- d_6 . A similar trend of downfield shift was observed in case of H_b as well and it exhibited the δ values of 8.05 ppm, 7.90 ppm, 7.95 ppm and 8.02 ppm, respectively in the solvent mixtures mentioned above (Figure 2e). On the other hand, in the ^{13}C NMR spectra, the carbonyl group of A-1 shifted from 185.7 ppm in neat acetone- d_6 to 178.8, 181.2, and 183.9 ppm in 6:1

D_2O /acetone- d_6 , 5:2 H_2O /acetone- d_6 , and 5:2 MeOH/acetone- d_6 , respectively (Figure 2f). These downfield shifts of the olefinic and aromatic protons (ortho to the carbonyl group of A-1) and carbonyl carbons are consistent with hydrogen-bonding interactions involving the naphthoquinone moiety, which further supports our working hypothesis.^{5b}”

Comment:

2. For the coumarin acylation, the product depicted is the α -acylated isomer (C108). Please clarify why the β -acylated regioisomer—expected on the basis of the mechanistic argument—is not observed.

Our response: We thank the reviewer for this question. The crude reaction mixture was analyzed in GC-MS. Neither of the α -acylated or β -acylated product was detected because coumarin failed to participate in the C–H HAT process under the standard conditions to generate the corresponding acyl radical for further reaction. The following outcome is now mentioned in the revised manuscript and highlighted in yellow.

“However, coumarin did not participate in the acylation reaction under the standard reaction conditions (remained unreacted), and neither of the anticipated products (C-108 or C-108’) was detected in GC-MS.”

Comment:

3. *N*-Phenyltetrahydropyrrole affords only trace amounts of C-136. Discuss whether this outcome arises from steric shielding, diminished H-bonding ability, or competing side reactions.

Our response: We thank the reviewer for asking this insightful question. The lower yield was observed, possibly due to the side reaction. In this particular case, the electron-rich amine center of *N*-phenyltetrahydropyrrole underwent photochemical oxidation (by SET) in the presence of Eosin Y and 1,4-naphthoquinone to deliver the corresponding iminium cation species, which experienced water

attack and successive oxidation to deliver *N*-phenyl pyrrolidone as the major product (*ChemSusChem*, **2019**, *12*, 2898 – 2910). Therefore, we have added the following line to the manuscript:

*“This is probably due to the photochemical oxidation of the electron-rich amine center of *N*-phenyltetrahydropyrrole, where SET is more favorable over HAT to generate an iminium cation under the standard reaction conditions. The corresponding iminium cation underwent a nucleophilic water attack and a successive oxidation to release *N*-phenylpyrrolidone (detected in GC-MS).²²”*

Comment:

4. Diaryl ketones are common triplet photosensitizers. When aryl aldehydes are used, the corresponding ketone products could conceivably act as internal sensitizers. Report Stern–Volmer quenching data, triplet-energy calculations, or sensitizer-free control reactions to exclude product-sensitized pathways.

Our response: We thank the reviewer for the valuable suggestion. We agree that diaryl ketones (formed upon quinone acylation) may act as photosensitizers. To negate the possibility of product-catalyzed reactions, we performed the standard reaction between 1,4-naphthoquinone (**A-1**) and benzaldehyde (**B-1**) in the absence of the photosensitizer Eosin Y but in the presence of **C-1** (5 mol%). We observed a trace amount of product formation in this case, and most of the starting materials remained unreacted, confirming that the reaction is not product-catalysed (analyzed by GC-MS). Furthermore, it is important to note that we obtain a dihydroxy diarylketone after the acylation reaction, which is highly susceptible to further oxidation and shows an absorption maximum at 404 nm for **C-1** (see SI, Fig. S3). We attempted to perform a Stern-Volmer quenching experiment for **C-1** and excited the molecule at 455 nm, but we did not observe any significant emission peak for the product. Afterwards, we tried to excite **C-1** at 400 nm, which provides an emission band. Next, for the quenching experiment we added 1,4-naphthoquinone (**A-1**) as the quencher. Unfortunately, we observed a messy emission band that contained many small peaks. This probably occurred because the product (**C-1**) is highly reactive, and photoexcitation of **C-1** at 400 nm in the presence of 1,4-naphthoquinone (**A-1**) led to its decomposition. These two experiments confirmed that the product-catalyzed reaction is unlikely under standard reaction conditions.

Reviewer #2 (Remarks to the Author):

The submitted manuscript by Konig and co-workers presents another application of on-water photochemistry in organic synthesis and follows up the original paper from the group in *Science*. The authors present the concept of influencing excited states by the water environment based on hydrogen bonds. This effect is manifested both on the substrate (benzoquinone and other quinones) and on the catalyst (EOSIN Y), and allows the activation of the substrate by energy transfer. Subsequently, acylation/alkylation of the substrate can occur, which is based on the abstraction of hydrogen from the

aldehyde or precursor of stable radicals. The authors document the effect of water on the absorption/fluorescence properties. Subsequently, they use an optimized methodology to prepare a series of acylated/alkylated quinones with various aldehydes or alkylating agents. Small-scale experiments are supplemented by experiments on a gram scale. The reactions tolerate various substrates and functional groups, with the exception of alkylating agents producing unstable radicals. Overall, I conclude that the work is clearly and comprehensibly written, most (see below for other comments) conclusions are clearly supported by measured data. Since this is a new and rapidly developing topic and the article, even in its current state, provides a guarantee (after minor edits) of an important message of interest to a wide range of readers, I recommend accepting the article for publication in Nature Communication after minor revision.

Comments:

1. Authors claim (see abstract) that water surface reduces triplet state energy of quinone substrate and increases triplet state energy of Eosin Y. On my own, it can be hardly concluded from data reported. Authors measured effect of water on absorption and fluorescence emission spectra but it can show only effect on singlet excited state. Of course, also triplet state energy might be changed but there is no direct evidence.

Our response: We thank the reviewer for this insightful feedback. We agree that the fluorescence emission spectra of any substance provide information only about the singlet state. The experimentally observed bathochromic shifts in the absorption and emission of **A-1** in the presence of water were confirmed by our computational studies, which suggest stabilization of the photoexcited states (both the singlet and triplet) in the aqueous environment (see SI, Fig S7-S9 for detailed calculations). However, the effect of water on the electronic structure of Eosin Y is more complex, as it may affect tautomeric equilibria and their individual electronics. Therefore, the following lines were added to the manuscript.

*“The experimentally observed bathochromic shift of absorption and emission of **A-1** in the presence of water was confirmed by the computational studies, which suggest the stabilization of the photo-excited states (both the singlet and triplet) for **A-1** in the aqueous environment (see SI, Fig S7-S9 for detailed calculations). However, the effect of water on the electronic structure of Eosin Y is more complex, as it may affect tautomeric equilibria and their individual electronics. A detailed evaluation goes beyond the scope of this work.²⁰”*

Additionally, the manuscript was modified accordingly and highlighted in yellow in the revised version.

Comments:

2. Regarding substrate scope of alkylation (Table 6), ethylbenzene derivative is missing. Only toluene and then trisubstituted derivatives were proven. It would be interesting to see whether switch to secondary derivative giving secondary radical is enough to provide the reaction.

Our response: We thank the reviewer for this interesting suggestion. Other than toluene, we performed the alkylation reaction in the presence of some secondary derivatives (C-124 to C-127), which shows that secondary carbon-centered radicals can participate in the reaction. However, for a substrate like ethylbenzene, we observed a different product. Further studies to investigate this different reactivity are currently ongoing in our lab; results on this will be reported in a future publication.

Comments:

3. Blue point is missing in legend in Figure 2D. Note, description A...D is not shown.

Our response: We thank the reviewer for pointing out these missing points. Figure 2D is now revised, and the description for 2A to 2D is now mentioned more elaborately and highlighted in yellow in the revised manuscript.

Comments:

4. Only traces of product are given in Table 2 for C23. But it is not commented accordingly in the text.

Our response: The outcome for C-23 is now mentioned in the revised manuscript and highlighted in yellow.

Comments:

5. Naphthaquinone should be naphthoquinone on page 7 down.

Our response: We apologise for the mistake. The typo is corrected in the revised manuscript and highlighted in yellow.

Comments:

6. Figure 3a is missing in the main text.

Our response: Figure 3 is now cited in the revised manuscript and highlighted in yellow.

Comments:

7. I suggest catalytic role of EY should be better shown in Figure 3. By excitation and ISC, triplet excited state is formed which likely provides EnT.

Our response: We thank this reviewer for the suggestion. The mechanism has been modified to more clearly highlight the role of Eosin Y.

Reviewer #3 (Remarks to the Author):

This manuscript describes a photo-induced, redox-neutral Friedel-Craft acylation and alkylation of quinones under water-oil interfaces in the presence of Eosin Y as photosensitizer. The method operates under transition-metal-free conditions, shows high substrate adaptability, and gives access to functionalized quinone products of potential synthetic value. The reaction design is conceptually appealing, and the operational simplicity makes it attractive. I recommend acceptance of this work for publication, upon addressing the following minor issues:

Comments:

1. In Table 2, the reason for lower yields for substrates like C-23, C-53 was not discussed. Did the author observe byproducts?

Our response: The lower yield for C-23 is now discussed in the revised manuscript and highlighted in yellow. The reason could be a different stereo-electronic factor operating in ortho-substituted aryl aldehydes bearing an electron-withdrawing group. However, for C-53, we observed an isolated yield of 76%, with the remaining starting materials unreacted.

Comments:

2. The authors should note that C-134 and C-135/C-136 are not the same class of compounds. I support the statement that C-134 can generate an α -amino radical via light-mediated HAT. However, applying the same description to C-135 and C-136 — that they also generate an α -amino radical via light-mediated HAT — may not be appropriate. For C-135 and C-136, a single-electron transfer (SET) pathway is likely more favorable than HAT.

In addition, for C-136, no obvious product was observed. What might be the possible reasons for this outcome? Is it necessary to further examine the reactivity of the N-phenylpiperidine substrate? For this type of substrate, the authors may need to discuss the mechanism of α -amino radical generation separately.

Our response: We thank the reviewer for the insightful feedback. We agree on the argument that for substrates like *N*-phenyltetrahydropyrrole, photochemical oxidation of the amine center by SET is more likely than C–H activation under the standard reaction conditions. The photo-oxidation of the electron-rich nitrogen center produced an iminium cation under the standard reaction conditions. The corresponding iminium cation underwent a nucleophilic attack by water, followed by oxidation, to release *N*-phenylpyrrolidone (detected by GC-MS) as the major product. The outcome is mentioned in the revised manuscript and highlighted in yellow. Surprisingly, for the substrate like *N*-phenylmorpholine (C-135), we could not detect any oxidation product, and the excess starting materials

remained unreacted. However, the alternative mechanistic route for this kind of reaction is now mentioned in the revised manuscript and highlighted in yellow.

“This is probably due to the photochemical oxidation of the electron-rich amine center of N-phenyltetrahydropyrrole, where SET is more favorable over HAT to generate an iminium cation under the standard reaction conditions. The corresponding iminium cation underwent a nucleophilic water attack and a successive oxidation to release N-phenylpyrrolidone (detected in GC-MS).²²”

Comments:

3. The scope of the reaction may be further illuminated by two key references (Angew. Chem. Int. Ed. 2016, 55, 4040; Chem. Sci. 2016, 7, 7002).

Our response: We thank the reviewer for the suggestion. The mentioned references are now cited in the revised manuscript and highlighted in yellow.

Comments:

4. Please verify that all citations and labels in Scheme 1 correspond accurately to their sources in the text and reference list.

Our response: We apologise for the mistake. All the citations in Scheme 1 are now labelled properly and highlighted in yellow in the revised manuscript.